# Effect of Agitation and Temporary Immersion on Growth and Synthesis of Antibacterial Phenolic Compounds in Genus *Drosera*

**DOI:** 10.3390/biom14091132

**Published:** 2024-09-07

**Authors:** Wojciech Makowski, Kinga Mrzygłód, Agnieszka Szopa, Paweł Kubica, Marta Krychowiak-Maśnicka, Krzysztof Michał Tokarz, Barbara Tokarz, Iga Ryngwelska, Ewa Paluszkiewicz, Aleksandra Królicka

**Affiliations:** 1Department of Botany, Physiology and Plant Protection, Faculty of Biotechnology and Horticulture, University of Agriculture in Krakow, 29 Listopada 54, 31-425 Krakow, Poland; salome269.40@gmail.com (K.M.); krzysztof.tokarz@urk.edu.pl (K.M.T.); barbara.tokarz@urk.edu.pl (B.T.); iga.ryngwelska788@gmail.com (I.R.); 2Department of Pharmaceutical Botany, Collegium Medicum, Jagiellonian University, Medyczna 9, 30-688 Krakow, Poland; a.szopa@uj.edu.pl (A.S.); p.kubica@uj.edu.pl (P.K.); 3Laboratory of Biologically Active Compounds, Intercollegiate Faculty of Biotechnology UG and MUG, University of Gdansk, Abrahama 58, 80-307 Gdansk, Poland; marta.krychowiak@ug.edu.pl; 4Faculty of Chemistry, Gdansk University of Technology, Narutowicza 11/12, 80-233 Gdansk, Poland; ewapalus@pg.edu.pl

**Keywords:** in vitro, carnivorous plants, 1,4-naphthoquinones, synthesis of ramentaceone, flavonoids, biological activity

## Abstract

Sundews (*Drosera* sp.) are the source of biologically active secondary metabolites: phenolic acids, flavonoids, and 1,4-naphtoquinones. Because obtaining them from the natural environment is impossible (rare and endangered species), in this study modifications of traditional tissue cultures grown in solid medium (SM), such as agitated cultures (ACs) (cultures in liquid medium with rotary shaking) and temporary immersion bioreactors Plantform^TM^ (TIB), were used for multiplication of four sundew species: *Drosera peltata*, *Drosera indica*, *Drosera regia*, and *Drosera binata*, with simultaneously effective synthesis of biologically active phenolic compounds. Each species cultivated on SM, AC, and TIB was tested for biomass accumulation, the content of total phenols and selected phenolic derivative concentrations (DAD-HPLC), the productivity on of phenolic compounds, as well as its antibacterial activity against two human pathogens: *Staphylococcus aureus* and *Escherichia coli*. The results showed that the type of culture should be selected for each species separately. Phytochemical analyses showed that the synthesis of secondary metabolites from the groups of phenolic acids, flavonoids, and 1,4-naphthoquinones can be increased by modifying the cultivation conditions. *D. regia* turned out to be the richest in phenolic compounds, including 1,4-naphtoquinones: plumbagin and ramentaceone. Extracts from *D. indica* and *D. regia* tissue showed strong antibacterial activity against both pathogens. It has also been shown that the growth conditions of sundews can modify the level of secondary metabolites, and thus, their biological activity.

## 1. Introduction

Carnivorous plants are known as “green predators” because of their ability to attract, capture, and digest small animals (mostly insects). Usually, carnivorous plants occur in poor habitats, where availability of nutrients is very limited. Carnivory allows them to absorb organic compounds (mostly nitrogen) from their prey’s body and supplement deficiency of key elements [1]. This unique adaptation to the natural environment implies specific features of carnivorous plant physiology. Researchers have proved that carnivorous syndrome in plants is closely connected with plants’ defense systems [2]. That is why most plants with carnivory syndrome have, among others, a natural ability to produce and accumulate in their tissues large amounts of various secondary metabolites. Such chemicals protect leaf traps during the digestion process and are involved in defense mechanisms against environmental stress [3]. Secondary metabolites in carnivorous plants may be accumulated as protective metabolites since they occur in habitats exposed to full sun [4]. Accumulated in epidermal cells, secondary metabolites can act as a screen against excess light and/or ultraviolet radiation with very high energy [5]. Moreover, secondary compounds neutralize reactive oxygen species (ROS), produced as a consequence of stress or during the digestive process in leaf traps [6].

The carnivorous genus *Drosera* sp. (sundew) is represented by nearly 250 species. It belongs, together with two monotypic genera, *Dionaea muscipula* and *Aldrovanda vesiculosa*, to the Droseraceae family [7]. Plants from the family Droseraceae have been used in folk medicine for centuries, because of various healing properties [8]. Nowadays, research focused on medical plants has shown that the antibacterial, antifungal, anti-inflammatory, antispasmodic, hepatoprotective, and anticancer properties of these plants result from the presence of various derivatives of phenolic compounds in their tissue [9]. The demand for biologically active plant metabolites is increasing due to the growing problem of antibiotic resistance among pathogenic bacteria [10]. Plants from the Droseraceae family synthesize various polyphenols including rare 1,4-naphtoquinones, known as having very potent biological activity [11]. The unique and rich composition of these metabolites in carnivorous plants’ organs makes them a potential source of medical secondary compounds for the industry [10]. However, data from qualitative and quantitative phytochemical analyses of selected *Drosera* species are very poor.

The limitation for the industrial use of carnivorous plant biomass is the fact that most of them are rare and endangered species. The International Union for the Conservation of Nature lists carnivorous plants as threatened (vulnerable, endangered, or critically endangered) [12]. Moreover, plant material acquired for green chemical purposes needs to be safe (free from contamination) and have a stable phenotype with repeatable features [13]. Furthermore, the accumulation level of secondary metabolites in plants in the natural environment is easily reduced due to unstable environmental conditions [14]. Therefore, tissue culture technology seems to be a good tool for the cultivation of sundews independent from environmental resources. This technology is a useful mass propagation method for the rapid multiplication of medical plants with a high content of target compounds [13]. Many carnivorous species have been cultivated in in vitro conditions [15]. Nevertheless, optimization of the growth conditions and modifications to traditional cultures cultivated on agar-solidified media may intensify biomass growth and the accumulation of secondary metabolites. Agitated cultures (tissue cultures in liquid medium with rotary shaking) (ACs) and temporary immersion bioreactors (TIBs) are both interesting tools to produce plant biomass with a high concentration of secondary metabolites [16]. In our previous research. we showed agitated cultures of *D. muscipula* to be a promising source of phenolic compounds [17]. Also, other authors have studied agitated cultures for medical plant propagation and shown a stimulation effect on growth and secondary metabolite production [18]. On the other hand, temporary immersion bioreactors were reported to be a useful tool for the scale-up of medical plant cultivation. It was proved that some species cultivated in bioreactors achieved higher concentrations of secondary metabolites [19]. Such equipment consists of a vessel and an automatic aeration and venting system that allows the liquid nutrient to move up and down. Thanks to the large volume of the growing vessel and periodic flooding of the plant root system, plant biomass can propagate quickly and efficiently [19].

In this research, four species of the genus *Drosera* were selected: *D. indica*, *D. peltata*, *D. regia*, and *D. binata* based on preliminary phytochemistry analysis (very rich in phenolic compounds). All four species synthesize plumbagin as the dominant 1,4-naphtoquinone, known for significant biological activity. Moreover, two species, *D. indica* and *D. regia*, produce the other rare 1,4-naphtoquinone with broad application potential in industry: ramentaceone—an isomer of plumbagin [9]. To be able to analyze the content of ramentaceone in the tissues of sundews, a ramentaceone standard was specially synthesized for this research.

*D. indica* is tropical sundew from Africa, southeast Asia, and Australia [12]. *D. peltata* occurs in South Korea [20], *D. binata* in New Zealand and Australia [21], while *D. regia* is an endemic species in South Africa [22]. They inhabit different ecological niches, and therefore, may have different requirements during cultivation. So, we hypothesized that *Drosera* species may demonstrate different strategies to cope with the applied growing conditions. For the first time, liquid cultures and bioreactors were tested for their suitability to produce biomass of these species and the secondary metabolites. Carnivorous plants are naturally connected with very wet habitats [17]. We believe that liquid medium conditions will enhance the growth of the examined plants. Moreover, sundew’s leaf traps are very sensitive to mechanical stimulation. So, the other hypothesis assumes that shaking of the liquid cultures will induce the defense mechanisms and lead to increased accumulation of phenolic compounds. On the other hand, temporary flooding in bioreactors may induce short hypoxia stress, what may increase synthesis of defensive secondary metabolites [13].

The main aim of this research was to check if modifications of traditional tissue cultures in agar-solidified medium, such as agitated cultures and temporary immersion bioreactors Plantform^TM^, are a good method for the effective multiplication of sundew plants with high concentrations of phenolic metabolites. In addition, extracts obtained from carnivorous plants was tested for their bactericidal activity to verify whether changes in growing conditions affect their biological activity.

## 2. Materials and Methods

### 2.1. Plant Material 

In the study, the previously established in vitro cultures of the sundews *D. peltata*, *D. indica*, *D. regia*, and *D. binata* from the collection of tissue cultures of the Laboratory of Biologically Active Compounds, Intercollegiate Faculty of Biotechnology UG and MUG, University of Gdansk, Poland, were used as a source of plant material. Cultures of whole plants for each *Drosera* species were maintained in 250 mL Erlenmeyer flasks on ½ Murashige and Skoog (½ MS) medium [23] solidified with 0.8% agar, without growth regulators, containing 3% sucrose, with pH 5.5 (adjusted prior to autoclaving). Plants were cultivated at 21 ± 2 °C, under white fluorescence light characterized by a photosynthetic photon flux density (PPFD) of 70 µmol × m^−2^ × s^−1^ and a photoperiod of 16 h/8 h light/dark cycle.

### 2.2. Examined In Vitro Culture Systems

#### 2.2.1. Cultures in Solid Medium 

Whole-plant cultures of each sundew species were grown in 250 mL Erlenmeyer flasks in solid medium (SM). For the experiment, 1 g of rooted plants per flask was used. Sundews were placed in 50 mL of ½ MS medium [23] solidified with 0.8% agar, without growth regulators, containing 3% sucrose, with pH 5.5 (adjusted prior to autoclaving). Biomass was collected after 6 weeks of growth cycles. Tissue cultures of *Drosera* plants in SM were cultivated at 21 ± 2 °C under white fluorescence light with a PPFD of 70 µmol × m^−2^ × s^−1^ and a photoperiod of 16 h/8 h light/dark cycle. The experiment consisted of 10 repetitions for each plant species (*n* = 10).

#### 2.2.2. Agitated Culture

The agitated culture (AC) of all the carnivorous plant species was carried out in 250 mL Erlenmeyer flasks. For the experiment, 1 g of rooted plants per flask was used. Plants were placed in 50 mL of ½ MS liquid medium [23], without growth regulators, containing 3% sucrose, with pH 5.5 (adjusted prior to autoclaving). Flasks were placed on a rotary shaker (Phoenix RS-LS 20, DanLab, Białystok, Poland) with continuous mode at 140 rpm. Biomass was collected after 6 weeks of growth cycles. ACs of *Drosera* plants were cultivated at 21 ± 2 °C under white fluorescence light with a PPFD of 70 µmol × m^−2^ × s^−1^ and a photoperiod of 16 h/8 h light/dark cycle. The experiment consisted of 10 repetitions for each species (*n* = 10).

#### 2.2.3. Cultures in Temporary Immersion Bioreactors 

Cultures of four *Drosera* species in temporary immersion bioreactors (TIBs) were cultivated in a Plantform^TM^ (Plant Form, Hjärup, Sweden). The immersion and aeration period in the bioreactors were programmed in the following cycle: 30 min of immersion, 20 min of gravitational fall of the medium, and 10 min of aeration. To start the experiment in the bioreactors, 10 g of whole, rooted plants was put into one container. One bioreactor contained 500 mL of liquid ½ MS medium [23], without growth regulators, containing 3% sucrose, with pH 5.5 (adjusted prior to autoclaving). Cultures of sundews in TIBs were cultivated at 21 ± 2 °C under white fluorescence light with a PPFD of 70 µmol × m^−2^ × s^−1^ and a photoperiod of 16 h/8 h light/dark cycle. Biomass samples from TIB cultures were collected after 6 weeks of growth cycles. The experiment consisted of 5 repetitions for each *Drosera* species (*n* = 5).

### 2.3. Growth Index and Dry Weight Estimation

Estimation of growth index (GI) was performed according to Tokarz et al. [1]. *D. peltata*, *D. indica*, *D. regia*, and *D. binata* cultivated in SM, LM, and TIB were harvested and weighed immediately. The GI was calculated according to the following formula: GI [%] = (FW_2_ − FW_1_)/FW_2_ × 100, where FW_1_ is the fresh weight of the plants at the beginning of the experiment and FW_2_ is the final fresh weight. To determine dry weight (DW) accumulation, the plants were freeze-dried for 48 h, and weighed. The DW content in the plant tissue was calculated according to the following formula: DW [%] = DW_2_ × 100/FW_2_, where DW_2_ is the dry weight after freeze-drying [1]. The freeze-dried plant tissue, separate for each genus, was homogenized to a powder and stored at −20 °C for further analysis.

### 2.4. Phytochemical Analysis

#### 2.4.1. Extraction Procedure

Plant extracts were prepared according to the procedure described by Makowski et al. [13], with modifications. For the preparation of methanolic extracts, 150 mg of homogenized lyophilized tissue powder was weighed out separately for each *Drosera* species, from each tested system. The tissue samples were subjected twice to extraction with 9 mL of 80% methanol (HPLC-grade) by sonication in an ultrasonic bath (POLSONIC 2, POLSONIC Palczyński Sp. J., Warszawa, Poland) for 30 min. Samples for each research object (each sundew from each condition) were prepared in 5 repetitions. The obtained extracts were filtered through sterilizing syringe filters (0.22 μm, Millex^®^GP, Millipore, Burlington, MA, USA) prior to HPLC analysis. 

#### 2.4.2. High-Pressure Liquid Chromatography

To estimate the accumulation of phenolic compounds in plant tissue, high-pressure liquid chromatography with a diode array detector (DAD-HPLC) was used. The quantitative analyses of selected secondary metabolites in the extracts were conducted with a validated method, using an apparatus of Merck-Hitachi (LaChrom Elite, Merck, Darmstadt, Germany) with a DAD L-2455 detector and on a Purospher RP-18 (250 × 4 mm; 5 μm, Merck, Darmstadt, Germany) column [24]. The flow rate was 1 mL × min^−1^, the temperature was set to 25 °C, and the injection volume was 10 μL. The detection wavelength was set to 254 nm. The mobile phase consisted of A—methanol, 0.5% acetic acid (1:4), and B—methanol (*v*/*v*). The gradient program was as follows: 0–20 min, 0% B; 20–35 min, 0–20% B; 35–45 min, 20–30% B; 45–55 min, 30–40% B; 55–60 min, 40–50% B; 60–65 min, 50–75% B; and 65–70 min, 75–100% B, with a hold time of 15 min. Identification was performed by comparison to the retention times and UV spectra of phenolic acids, flavonoids, and plumbagin standards (acquired from Sigma-Aldrich Co., Darmstadt, Germany) and a ramentaceone standard obtained by chemical synthesis, described below. The quantification was performed based on the calibration curve method. Samples were prepared and analyzed in five replications. The results are expressed in mg × 100 g^−1^ DW. 

### 2.5. Chemical Synthesis of Ramentaceone (5-Hydroxy-7-methyl-1,4-naphthoquinone)

The synthesis of 5-hydroxy-7-methyl-1,4-naphthoquinone (ramentaceone) was carried out in two stages. The first stage was the preparation of the 8-chloro-5-hydroxy-7-methyl-1,4-naphthoquinone derivative (a method known from publications; the efficiency of the first stage was 23%) [25,26], followed by the reduction of the chlorine derivative to the expected product.

A mass of 91 mg (0.4 mmol) of 8-chloro-5-hydroxy-7-methyl-1,4-naphthoquinone was dissolved in 30 mL of methanol and 14 mg of palladium of carbon (Aldrich—MilliporeSigma, St. Louis, MO, USA) was added. The reaction mixture was hydrogenated for 2.5 h. After this time, the catalyst was filtered into a mixture of FeCl_3_ solution (2.5 g in 20 mL H_2_O). The solution was extracted with methylene chloride twice. The combined organic layers were evaporated under reduced pressure. The compound was purified by column chromatography (silica gel was suspended in petroleum ether). Eluent petroleum ether:ethyl acetate, 15:1. After evaporation of the fractions containing the product, the compound was subjected to structural analysis: reaction yield 73% (56 mg), m.p. 122–123 °C.

^1^H NMR (500 MHz, CDCl3) δppm: 11.88 (s, 1H, 5-OH), 7.45 (s, 1H, H-8), 7.10 (1H, s, H-6), 6.93 (s, 2H, H-2, H-3), 2.45 (s, 3H, CH_3_); ESI-MS C_11_H_8_O_3_—188.18 g/mol, mass [M^−^] 187.0 (Appendix A).

The structure of the synthesized compound was confirmed using spectroscopic methods: mass spectrometry ESI-MS and proton magnetic resonance. The purity was ascertained using thin-layer chromatography (TLC). Proton nuclear magnetic resonance (^1^H NMR) spectra were recorded on a Varian VXR-S spectrometer operating at 500 MHz. Melting points were determined on a Stuart SMP30 capillary apparatus and were uncorrected. Mass spectra were recorded using an Agilent 6470A triple quadrupole liquid chromatography–tandem mass spectroscopy (LC/MS) system with an electrospray ionization source (ESI) in SCAN mode (Appendix A). Samples were prepared as 1 μg/mL solutions in methanol and were supplied in 1 μL aliquots to the mass spectrometer in the mixture of acetonitrile:water:formic acid (38:57:5 *v*/*v*/*v*) at a flow rate of 500 μL/min. Chemical shifts are reported as δ units in ppm downfield from internal tetramethylsilane. The NMR abbreviations used are as follows: br.s—broad signal, s—singlet, d—doublet, dd—doublet of doublets, t—triplet, k—quartet, and m—multiple. The results of the elemental analyses for individual elements fit within ± 0.4% of the theoretical values.

### 2.6. Calculation of the Sum of Phenolic Compounds

The sum of phenolic compounds (SPCs) was calculated by summing up the content of all the tested phenolic derivatives: gallic acid, protocatechic acid, chlorogenic acid, caffeic acid, ferulic acid, ellagic acid, hyperoside, myricetin, quercetin, kaempferol, plumbagin, and ramentaceone. The results are expressed as mg × 100 g^−1^ DW.

### 2.7. Calculation of the Productivity of Phenolic Compounds

The productivity of phenolic compounds (PROD), phenolic acids, flavonoids, and naphtoquinones, in each culture type of each sundew was calculated according to the formula PROD [mg × 100 mL^−1^ of medium] = A × B/C, where A is the sum in mg of phenolic acids, flavonoids, or naphtoquinones in plant tissue after 6 weeks of growth per 1 g DW; B is the quantity of g DW in one flask/bioreactor; and C is 100 mL of medium in the flask/bioreactor.

### 2.8. Antibacterial Activity of Plant-Derived Extracts

To evaluate the bactericidal properties of the examined plants, minimal bactericidal concentration (MBC) was determined with the broth microdilution method for each tested plant extract [17]. MBCs were evaluated against the human-pathogenic bacteria *Staphylococcus aureus* ATCC 25923 and *Escherichia coli* ATCC 25922, obtained from the IFB UG and MUG Poland. The bacteria were cultivated on BHI medium (overnight, 37 °C). Freeze-dried plant tissue (100 mg) was extracted in 80% methanol [13]. Extracts were evaporated and resuspended in methanol before application into wells of the 96-well plate. To remove toxic methanol, extracts were evaporated in the wells. The residues were suspended in 100 μL of BHI medium, and 10 μL aliquots of bacterial suspension in BHI (10^5^ CFU × mL^−1^) were added to each well. Plates were incubated overnight. To establish the MBC value, 100 μL from each well that showed no visible growth of bacteria was plated out on a BHI agar plate for 24 h of incubation at 37 °C. The MBC was defined as the lowest concentration of the extract that reduced the inoculum by 99.9% within 24 h. 

### 2.9. Statistical Analyses

Two-way analysis of variance (ANOVA) was used to determine significant differences between means (Tukey test at *p* < 0.05 level). STATISTICA 12.0 (StatSoft Inc., Tulsa, OK, USA) was used to carry out statistical analyses. Homogeneous groups in charts/tables are marked with letters.

## 3. Results

### 3.1. Morphological Observations of Drosera Tissue Cultures

Observations of plants’ morphology, viability, and potential stress symptoms were conducted at the end of cultivation. All four studied species had similar morphology between the tested conditions. The plants were healthy, without macroscopic stress symptoms (Figure 1).

The growth index (GI) and dry weight (DW) of plant tissue were determined to evaluate the growth of sundews in the applied experimental conditions (Table 1). However, we observed species-specific variation in the growth of the tested sundews in response to the applied conditions. Agitation of the medium increased the growth of *D. peltata* and *D. indica* by 28 and 31%, respectively, compared to the control. Temporary immersion in the bioreactor, on the other hand, increased the growth of *D. indica* and *D. regia* by 39% and 24%, respectively. For *D. binata*, the control conditions, i.e., culture in standard agar-solidified media, appeared to be the optimal conditions to stimulate growth of plant tissue.

The presented results indicate that, compared to solid medium, the agitated cultures had a significantly lower DW content, while the bioreactor cultures had a higher one. The bioreactors increased the accumulation of DW in *D. peltata* and *D. regia* by 1.4- and 1.6-fold, while the agitated cultures decreased DW in *D. indica* by 50%.

### 3.2. Sum of Phenolic Compounds in Sundew Tissue Cultures

The calculation of the sum of the estimated phenolic derivatives allowed the study of the differences in polyphenolic content depending on the sundew species and the cultivation conditions used. Among the tested sundews, *D. regia* and *D. binata* turned out to be the richest in the studied secondary metabolites. The highest accumulation level of polyphenols was obtained in solid medium and bioreactor cultures of *D. regia*, reaching 7833 and 8023 mg × 100 g DW of phenolic compounds, respectively. Agitated cultures of *D. indica* and *D. binata* accumulated higher concentrations of phenolics than other types of cultivation conditions tested for these two species (Figure 2).

### 3.3. Accumulation of Phenolic Derivatives in Sundew Tissue Cultures

DAD-HPLC extracts from sundew tissues were analyzed and numerous compounds with polyphenol structures were detected. Six phenolic acids (Table 2), four flavonoids, and two 1,4-napthoquinones (Table 3) were quantified. The amounts of the phenolic derivatives depended on the sundew species and were affected by the type of tissue culture. Agitation of medium during cultivation stimulated accumulation of phenolic acids in the examined sundew species. The highest levels of gallic and chlorogenic acid, i.e., 44.8 and 38.6 mg × 100 g^−1^ DW, respectively, were found in the agitated cultures of *D. peltata*. The same conditions led to the most effective synthesis of protocatechuic acid in *D. indica* at the level of 16.9 mg × 100 g^−1^ DW. The agitated cultures of *D. regia* accumulated the highest levels of caffeic (36.1 mg × 100 g^−1^ DW), ferulic (241.5 mg × 100 g^−1^ DW), and ellagic acids (142.2 mg × 100 g^−1^ DW) compared to the other studied culture conditions and sundew species (Table 2).

Quantitative analysis of flavonoids, i.e., hyperoside, myricetin, quercetin, and kaempferol, showed that regardless of growing conditions tissues of *D. regia* and *D. binata* accumulated significantly higher levels of flavonoids than tissues of *D. peltata* and *D. indica* (Table 3). The source of the highest concentration of hyperoside was cultures of *D. regia* in solid medium (1337.3 mg × 100 g^−1^ DW). In contrast, the highest concentrations of myricetin, i.e., 338.3 and 331.6 mg × 100 g^−1^ DW, were found in tissues of *D. regia* cultured in solid medium and exposed to temporary immersion in the bioreactor, respectively. The highest level of quercetin (between 908.3 mg × 100 g^−1^ DW and 982.0 mg × 100 g^−1^ DW) was found in extracts from *D. binata* cultivated using standard conditions, i.e., cultivation in solid medium. Tissues of *D. regia* and *D. binata* cultured in agitated medium accumulated the highest level of kaempferol, with concentrations of 141.8 and 123.0 mg × 100 g^−1^ DW (Table 3).

Among the phenolic compounds detected in the tissues of sundews, the dominant metabolite was plumbagin, which belongs to the class of 1,4-naphthoquinones. The other analyzed 1,4-naphthoquinone was an isomer of plumbagin, ramentaceone, the presence of which has been demonstrated only for *D. indica* and *D. regia. D. regia* cultivated in solid medium and the bioreactor showed the most significant potential for synthesis of 1,4-naphthoquinones, where plumbagin concentrations were 4069.7 and 4606.2 mg × 100 g^−1^ DW, and ramentaceone 1358.7 and 1268.0 mg × 100 g^−1^ DW, respectively (Table 3). 

### 3.4. Phenolic Compounds Productivity in Sundew Tissue Cultures

The potential productivity of phenolic acids, flavonoids, and naphthoquinones in tissue cultures of sundews were dependent on the applied cultivation conditions. The highest phenolic acid productivity level was obtained in solid medium cultures of *D. indica* and *D. regia*, as well as agitated cultures of *D. regia.* Moreover, agitated culture conditions led to increased production of phenolic acids in *D. peltata*, which increased 2.4-fold compared to solid medium cultures and 3.3-fold compared to bioreactor cultures. On the contrary, solid medium cultures of *D. binata* were more effective for phenolic acid productivity than modified culture conditions (Figure 3).

Similar to phenolic acids, we observed increased production of flavonoids in *D. peltata* in agitated cultures compared to solid medium and bioreactor cultures. Nevertheless, the most effective flavonoid productivity was obtained in solid medium cultures of *D. binata* (41.8 mg × 100 mL^−1^ medium). The same conditions led to increased productivity of flavonoids in *D. regia* compared to agitated and bioreactor cultures, achieving a level of 31.1 mg × 100 mL^−1^ medium (Figure 4).

In case of naphthoquinones the highest level of productivity was obtained for solid medium cultures of *D. regia* (87.2 mg × 100 mL^−1^ medium). The same conditions led to the increased production of naphthoquinone in *D. binata* (69.8 mg × 100 mL^−1^ medium) compared to cultures cultivated in modified conditions. Tissues of *D. peltata* and *D. indica*, in turn, accumulated higher levels of naphthoquinone when growing in solid and agitated medium compared to tissues cultivated in the bioreactor (Figure 5).

### 3.5. Bactericidal Properties of Sundew Tissue Cultures

The minimal bactericidal concentration (MBC) of plant tissue extracts was determined against two human-pathogenic bacteria to compare the antibacterial potential of secondary metabolites accumulated in sundew tissues cultivated under distinct conditions. *D. indica* and *D. regia* extracts had two times higher bactericidal properties against Gram-positive *S. aureus* compared to *D. peltata* and *D. binata*. Similarly, *D. indica* and *D. regia* were characterized by higher bactericidal activity against Gram-negative *E. coli* than the remaining tested sundews (Table 4).

The MBCs of extracts from sundews was only slightly affected by the condition of the tissue cultures. A 2-fold increase in bactericidal potential against *S. aureus* was observed only for the extract from *D. binata* cultured in solid medium compared to plant tissues cultured in agitated medium and in the bioreactor (Table 4). The bactericidal potential against *E. coli* was affected only for extracts from *D. peltata* and *D. indica* (Table 4). Agitation of the medium resulted in doubled MBCs of *D. indica* extracts and, conversely, halved MBCs of *D. peltata* (Table 4).

## 4. Discussion

The *Drosera* genus represents a group of plants very rich in polyphenol compounds characterized by biological activity [27]. Sundews disappeared from natural therapy because of natural conservation [28]. The latest research shows that plants from the Droseraceae family are a source of secondary metabolites with an exceptionally strong antibacterial effect [10]. Therefore, in times of increasing antibiotic resistance of pathogenic bacteria, they can be used to produce medicinal secondary metabolites. First, however, it is necessary to develop effective protocols for their micropropagation, which will make the cultivation independent of natural environmental resources and provide plant material rich in the sought-after plant metabolites.

Tissue culture technology may be a useful tool for the propagation of medical plant species for the synthesis of valuable secondary metabolites with broad biological activity [13]. In this study, new approaches to cultivation of four *Drosera* species were studied, where modifications of culture systems were applied. Both cultivation systems, agitated cultures (ACs) and temporary immersion bioreactors (TIBs), were characterized by better nutrient distribution and oxygen availability in the medium, compared to solid medium (SM). Moreover, in TIBs the level of ethylene is low, because of the aeration phase. Taken altogether, plants cultivated in such conditions may be characterized by increased growth [29]. In our study, analysis of sundews’ growth index showed that LM increased biomass accumulation in *D. peltata* and *D. indica* plants, up to 1.28- and 1.31-fold, respectively. Nevertheless, the same parameter decreased for *D. binata* in LM, compared to plants from SM. Also, AC systems led to a decrease in dry weight accumulation in *D. indica* plants. Previously, continuously shaking the medium was proved to be a better environment for the biomass accumulation of another Droseraceae plant—*D. muscipula*—than agar-solidified medium [17]. Moreover, the agitated culture system was shown to be more effective than traditional SM for cultivation of *Arnebia euchroma* and *Scutellaria alpina* [18,30]. The TIB showed growth of *D. regia*. This may relate to the fact that this sundew is one of the biggest in the world and the bioreactor does not limit its shape. We hypothesized that AC conditions are better for small sundews’ cultivation, like *D. peltata* and *D. indica*. Plants with a smaller size are not so exposed to intense mechanical stress (shaking the culture). However, for sundews with long/large leaves that touch the walls of the flask, the mechanical stimulus is constant and may lead to a reduced growth rate.

Phytochemical analysis with the use of DAD-HPLC allowed for the determination of the content of phenolic compounds in sundew tissues and the calculation of the sum of phenolic compounds. This parameter showed that *D. regia* cultivated in SM or TIB was characterized by the highest level of polyphenol accumulation. ACs decreased production of phenolic compounds in *D. regia.* A similar phenomenon was shown by Kawka et al. [31], where the level of verbascosides decreased in ACs of *Scutellaria lateriflora* L. compared to SM. Contrary, the second sundew in terms of total phenolic content—*D. binata*—showed increased phenolic accumulation in ACs, the same as *D. indica*. Mechanical stimulation during constant shaking of ACs may act as an elicitor for the sensitive leaves of carnivorous plants. Mechanical stimulus may lead to a stress reaction, which may manifest itself in an increase in the synthesis of secondary metabolites [17]. 

Phenolic acids are one of the dominant and common groups of polyphenols in sundews [9]. Due to their chemical structure, they perform several functions in plant organisms [32] and are also characterized by strong biological activity [17]. In this study, qualitative analysis of sundew extracts allowed for the estimation of the content of six phenolic acids: gallic, protocatechic, chlorogenic, caffeic, ferulic, and ellagic acids. Some of them were previously reported in *D. peltata*, *D. indica*, or *D. binata* [9], while none of them have been reported in *D. regia*. Our study showed that among the studied species of carnivorous plants, *D. regia* is the richest in phenolic acids. Nevertheless, cultivation conditions affected the accumulation of phenolic acids in sundews. The results of the analysis showed that the highest synthesis of phenolic acids takes place in AC conditions. ACs of *D. peltata* had the highest values of gallic and chlorogenic acids, ACs of *D. indica* accumulated the highest levels of protocatechic and ellagic acids, while in the *D. regia* ACs the highest concentrations of caffeic, ferulic, and ellagic acids (together with *D. indica*) were found. Szopa et al. [16] showed that ACs of *Schizandra chinensis* are an effective source of phenolic acids. ACs of *D. muscipula* accumulated more phenolic acids than SM cultures [17]. Another parameter that allows the determination of the efficiency of production of a given group of compounds in a specific time, based on the accumulation of dry matter and the volume of the medium, is productivity. The highest production of phenolic acids was found for SM cultures of *D. regia.* The same type of tissue culture increased phenolic acid productivity in *D. indica* and *D. binata* compared to ACs or TIBs. Only *D. peltata* ACs had increased productivity of phenolic acids compared to SM and TIBs. Of the eight phenolic acids analyzed in *Pontechium maculatum*, the productivity of three, isoferulic, isochlorogenic, and rosmarinic acids, was increased in ACs compared to SM [13]. Nevertheless, the highest phenolic acid productivity was obtained in TIBs [13]. 

In this study, the composition of sundew extracts was analyzed and four flavonoids, hyperoside, myricetin, quercetin, and kaempferol, were evaluated. Flavonoids are the group of secondary metabolites able to protect plants against UV stress (as screening metabolites), but also against fungal infections and other stresses [9]. Moreover, these compounds are the colorants of flowers and very active free-radical scavengers [32]. Analysis showed that the level of flavonoids is primarily related to the species. *D. regia* and *D. binata* naturally accumulate significantly larger concentrations of flavonoids than the other two sundews. Therefore, despite the influence of cultivation conditions on the level of these compounds in sundews, the highest amounts were accumulated in *D. regia* and *D. binata.* The highest level of hyperoside was found for the TIB culture of *D. regia.* In addition to the bioreactor, SM also induced the highest myricetin production in *D. regia* tissues. Regardless of the cultivation conditions, the highest quercetin content was produced by *D. binata*, while ACs of both *D. regia* and *D. binata* synthesized the highest levels of kaempferol. Similar to the case of phenolic acids, the most effective flavonoid productivity was found for *D. regia* and *D. binata* in SM, while AC increased flavonoid productivity in *D. peltata.* The content of flavonoids in *S. chinensis* decreased in AC compared to SM [18]. Kawka et al. [31] showed a lower total flavonoid accumulation in *S. lateriflora* from AC compared to SM.

The third group of secondary metabolites analyzed in the sundew cultures was 1,4-naphtoquinones. These rare metabolites are present in the whole Droseraceae family, with the dominant one usually being plumbagin [9]. Some species of sundews also synthesize ramentaceone, which is an isomer of plumbagin, in their tissues. Because of the broad biological activity of plumbagin and ramentaceone, they are a topic of interest for science and industry [3]. Due to the impossibility of purchasing ramentaceone as a standard on the commercial market, it was specially synthesized for the purposes of this research. Our study showed that each of the examined sundews synthesize plumbagin, but just two, *D. indica* and *D. regia*, also produce ramentaceone. The highest source of both 1,4-naphtoquinones was *D. regia*. Plumbagin accumulation was increased in TIB cultures of *D. regia*, while ramentaceone was increased in TIB and SM cultures. It is currently believed that *D. muscipula* is the carnivorous plant richest in plumbagin, reaching concentrations between 6000 and 7000 mg × 100 g^−1^ DW [33]. The results of this study showed *D. regia* to be a very rich source of 1,4-naphtoquinones for the first time. Naphthoquinone productivity was also affected by cultivation conditions. The highest value was obtained in *D. regia* SM cultures. SM cultures of *D. binata* were also characterized by increased naphthoquinone levels compared to ACs and TIBs. In our previous research, we demonstrated that the synthesis and production of shikonin (1,4-naphtoquinone) in *P. maculatum* is increased in TIB cultures [13]. Moreover, TIBs increased the plumbagin yield from *Drosera communis* compared to SM and stationary liquid cultures [34].

Because of the growing problem of antibiotic resistance in pathogenic bacteria, it is important to discover new sources of substances with antibacterial potential [10]. Our previous studies showed that *D. muscipula* is a potent source of secondary metabolites with strong bactericidal properties [17,33]. This research has focused on the possibility of using the tissues of the sundew as an antibacterial agent. The aim of this part of the study was to evaluate if each *Drosera* has potential for bacteria elimination and how species and/or cultivation conditions affect the biological activity of sundews. The most powerful against Gram-positive *Staphylococcus aureus* and Gram-negative *Escherichia coli* were extracts from *D. indica* and *D. regia*, because minimal bactericidal concentration (MBC) was obtained with a lower tissue quantity than in the cases of *D. peltata* and *D. binata.* Interestingly, SM and TIB cultures of *D. indica* had two times higher antibacterial power against *E. coli* than ACs. By contrast, ACs stimulated the bactericidal properties in *D. peltata* compared to SM and the TIB. Also, the SM culture of *D. binata* showed more antimicrobial potential than the AC or TIB. In the work of Gerschler et al. [35], the potential of European sundews to inhibit the formation of biofilms created by various antibiotic-resistant strains of *E. coli* was demonstrated. Based on their research, the authors suggest that the most powerful antibacterial metabolites in the sundew herb are flavonoids and 1,4-naphthoquinones. In our previous study on *D. muscipula,* we showed that phenolic acid concentrations in plant-derived extracts are also important to effectively combat bacteria [17]. The bactericidal capacity may be affected by elicitors, genetic transformation, or the cultivation conditions [33]. Based on the obtained results, we can conclude that *D. indica* and *D. regia* have very strong antibacterial activity, like the values obtained for *D. muscipula* [17,33]. An interesting phenomenon is the strong antibacterial potential of *D. indica* from SM and the TIB against Gram-negative *E. coli*. Gram-negative bacteria usually require more bactericidal compounds than Gram-positive to eliminate them due to the protein–lipid membrane covering their cell wall [33]. Considering that the MBC for pure plubmagin and ramentaceone is 50 μg/mL against *E. coli* and the content of both naphthoquinones in the effective bactericidal dose is 18.7 μg for SM and 22.7 μg for TIB, it can be concluded that the antibacterial activity of *D. regia* extracts does not depend only on the content of naphthoquinones. The secondary metabolites in this sundew extract act synergistically to enhance its antimicrobial activity. The results provide a strong basis for research into the use of sundew tissue cultures to produce antibacterial secondary metabolites.

## 5. Conclusions

Based on the obtained results, we can conclude that agitated cultures and temporary immersion bioreactors Plantform^TM^ may be useful tools for the fast and effective propagation of *Drosera* plants. However, due to the different morphology and sensitivity of sundews, it is necessary to select tools for individual species. Of the four carnivorous plant species studied, *D. regia* is the plant from the Droseraceae family richest in secondary metabolites that have been phytochemically characterized, providing the basis for research into its use in the production of biologically active phenolic compounds. The strong bactericidal properties of sundews are related to the presence of various polyphenols, but the most important compounds in the fight against bacteria seem to be 1,4-naphthoquinones. The enhanced antibacterial properties of *D. indica* and *D. regia* are probably related to the fact that, in addition to plumbagin, they also accumulate ramentaceone in their tissues.

## Figures and Tables

**Figure 1 biomolecules-14-01132-f001:**
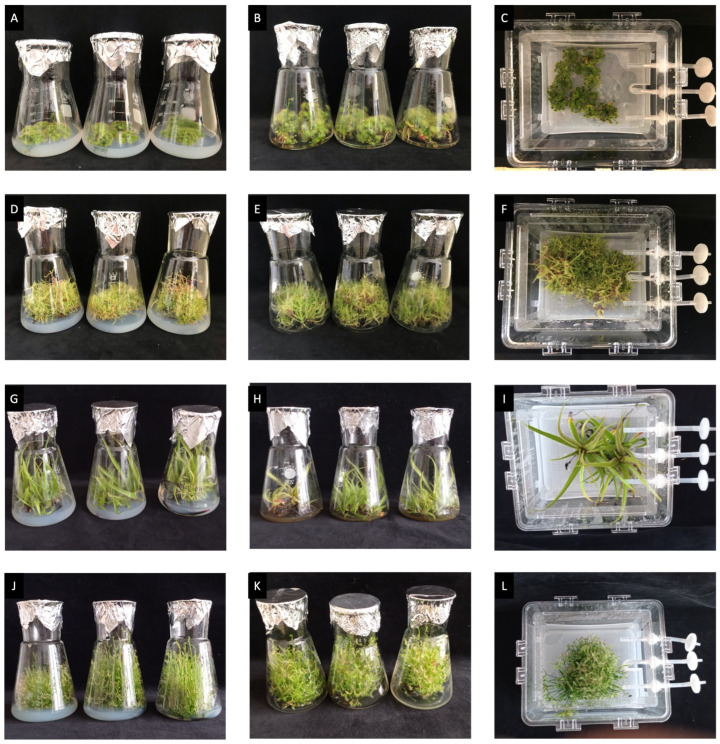
Plants after 6 weeks of cultivation: *D. peltata* (**A**–**C**); *D. indica* (**D**–**F**); *D. regia* (**G**–**I**); *D. binata* (**J**–**L**). (**A**,**D**,**G**,**J**)—solid medium; (**B**,**E**,**H**,**K**)—agitated culture; (**C**,**F**,**I**,**L**)—temporary immersion bioreactor Plantform^TM^.

**Figure 2 biomolecules-14-01132-f002:**
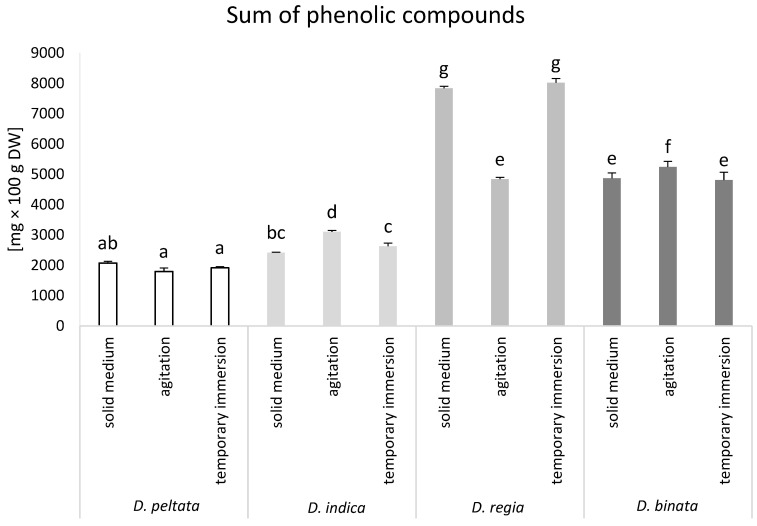
Sum of phenolic compounds (mg × 100 g^−1^ DW) in *D. peltata*, *D. indica*, *D. regia*, and *D. binata* cultivated in solid medium, agitated cultures, and temporary immersion bioreactor Plantform^TM^. Lower case letters indicate statistical significance of means according to two-way ANOVA, post hoc Tukey test at *p* < 0.05; the bar represents the standard deviation; DW—dry weight.

**Figure 3 biomolecules-14-01132-f003:**
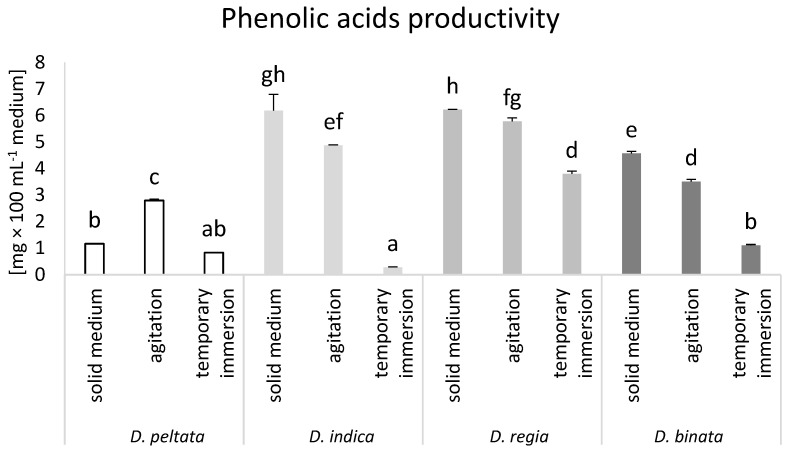
Phenolic acid productivity (mg × 100 mL^−1^ of medium) in *D. peltata*, *D. indica*, *D. regia*, and *D. binata* cultivated in solid medium, agitated cultures, and temporary immersion bioreactor Plantform^TM^. Lower case letters indicate statistical significance of means according to two-way ANOVA, post hoc Tukey test at *p* < 0.05; the bar represents the standard deviation.

**Figure 4 biomolecules-14-01132-f004:**
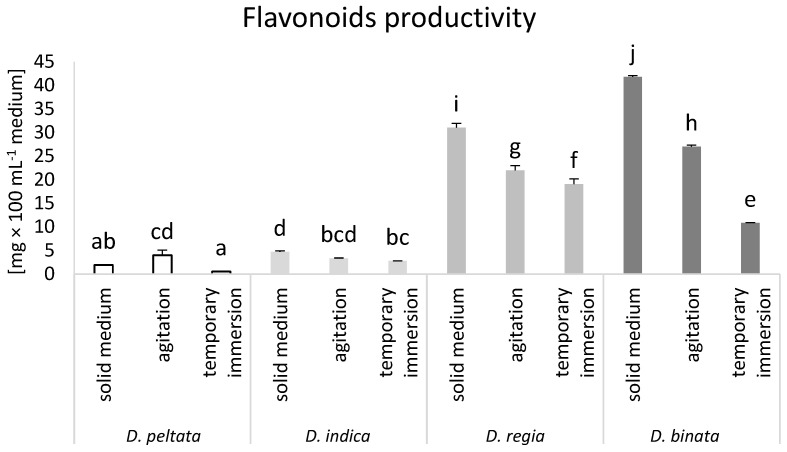
Flavonoid productivity (mg × 100 mL^−1^ of medium) in *D. peltata*, *D. indica*, *D. regia*, and *D. binata* cultivated in solid medium, agitated cultures, and temporary immersion bioreactor Plantform^TM^. Lower case letters indicate statistical significance of means according to two-way ANOVA, post hoc Tukey test at *p* < 0.05; the bar represents the standard deviation.

**Figure 5 biomolecules-14-01132-f005:**
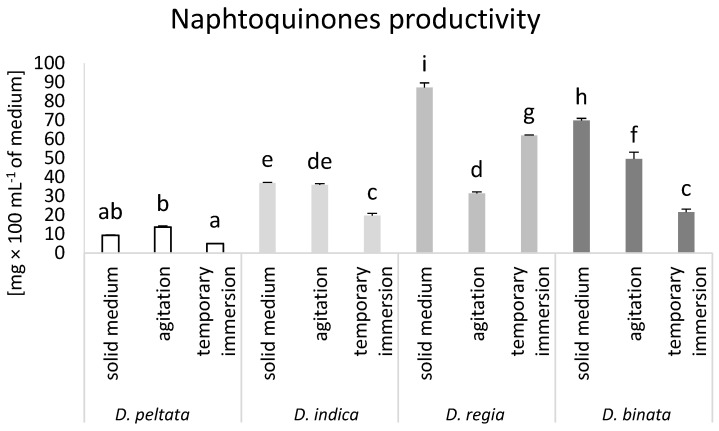
Naphthoquinone productivity (mg × 100 mL^−1^ of medium) in *D. peltata*, *D. indica*, *D. regia*, and *D. binata* cultivated in solid medium, agitated cultures, and temporary immersion bioreactor Plantform^TM^. Lower case letters indicate statistical significance of means according to two-way ANOVA, post hoc Tukey test at *p* < 0.05; the bar represents the standard deviation.

**Table 1 biomolecules-14-01132-t001:** Growth index (GI) and dry weight (DW) content (%) in *D. peltata*, *D. indica*, *D. regia*, and *D. binata* plants cultivated in solid medium, agitated cultures, and temporary immersion bioreactor Plantform^TM^. Different letters indicate statistical significance of means according to two-way ANOVA, post hoc Tukey test at *p* < 0.05; SD—standard deviation.

		Growth Index	Dry Weight
		% [±SD]	
*Drosera peltata*	Solid medium	61.3 ± 9.3 ^bc^	16.5 ± 1.8 ^c^
	Agitation	78.5 ± 2.2 ^e^	14.1 ± 1.6 ^bc^
	Temporary immersion	57.4 ± 8.3 ^ab^	23.2 ± 2.5 ^d^
*Drosera indica*	Solid medium	56.2 ± 7.0 ^ab^	12.4 ± 0.5 ^bc^
	Agitation	73.9 ± 4.1 ^de^	6.3 ± 0.4 ^a^
	Temporary immersion	78.4 ± 3.8 ^e^	17.5 ± 4.6 ^c^
*Drosera regia*	Solid medium	58.3 ± 6.7 ^ab^	16.0 ± 1.0 ^c^
	Agitation	58.9 ± 6.0 ^ab^	14.1 ± 2.1 ^bc^
	Temporary immersion	72.2 ± 0.3 ^de^	25.1 ± 4.8 ^d^
*Drosera binata*	Solid medium	63.9 ± 3.9 ^bcd^	7.5 ± 0.5 ^a^
	Agitation	51.3 ± 5.6 ^a^	6.9 ± 0.5 ^a^
	Temporary immersion	62.4 ± 0.5 ^abcd^	9.5 ± 1.6 ^ab^

**Table 2 biomolecules-14-01132-t002:** Accumulation of phenolic acids (mg × 100 g^−1^ DW) in *D. peltata*, *D. indica*, *D. regia*, and *D. binata* cultivated in solid medium, agitated cultures, and temporary immersion bioreactor Plantform^TM^. Different letters indicate statistical significance of means according to two-way ANOVA, post hoc Tukey test at *p* < 0.05; SD—standard deviation, DW—dry weight.

		Gallic Acid	Protocatechic Acid	Chlorogenic Acid	Caffeic Acid	Ferulic Acid	Ellagic Acid
		mg ×100 g^−1^ DW [±SD]					
*Drosera peltata*	Solid medium	39.5 ± 0.1 ^f^	2.5 ± 0.1 ^ab^	25.7 ± 0.1 ^f^	6.2 ± 0.2 ^ab^	20.9 ± 0.1 ^ab^	98.0 ± 8.5 ^d^
	Agitation	44.8 ± 0.1 ^g^	3.1 ± 0.1 ^b^	38.6 ± 0.2 ^g^	8.0 ± 0.2 ^abc^	36.1 ± 0.2 ^ab^	98.4 ± 7.6 ^d^
	Temporary immersion	28.6 ± 0.3 ^d^	11.0 ± 0.2 ^f^	9.0 ± 0.1 ^bc^	5.1 ± 0.1 ^a^	109.6 ± 3.5 ^c^	90.6 ± 0.9 ^cd^
*Drosera indica*	Solid medium	39.7 ± 0.3 ^f^	10.8 ± 0.1 ^ef^	12.7 ± 0.3 ^d^	8.3 ± 0.1 ^abc^	141.8 ± 9.4 ^cd^	93.4 ± 15.6 ^cd^
	Agitation	33.6 ± 0.2 ^e^	16.9 ± 0.1 ^h^	7.6 ± 0.1 ^b^	6.6 ± 0.1 ^ab^	157.7 ± 5.5 ^de^	123.5 ± 0.6 ^e^
	Temporary immersion	5.2 ± 0.2 ^a^	1.8 ± 0.1 ^a^	1.8 ± 0.6 ^a^	2.5 ± 0.2 ^a^	9.3 ± 0.1 ^a^	11.4 ± 0.4 ^a^
*Drosera regia*	Solid medium	33.5 ± 1.1 ^e^	9.8 ± 0.2 ^d^	16.4 ± 0.2 ^e^	32.0 ± 11.0 ^ef^	231.8 ± 37.9 ^f^	89.7 ± 0.3 ^cd^
	Agitation	26.7 ± 0.3 ^bc^	7.2 ± 0.4 ^c^	7.3 ± 0.3 ^b^	36.1 ± 0.9 ^g^	241.5 ± 18.4 ^f^	142.2 ± 2.3 ^e^
	Temporary immersion	27.9 ± 0.3 ^cd^	9.9 ± 0.3 ^de^	11.2 ± 2.4 ^cd^	17.6 ± 1.3 ^cd^	180.4 ± 11.6 ^e^	102.7 ± 10.7 ^d^
*Drosera binata*	Solid medium	27.7 ± 0.9 ^cd^	9.8 ± 0.7 ^d^	11.5 ± 1.8 ^cd^	16.0 ± 3.3 ^bcd^	45.9 ± 1.8 ^ab^	77.8 ± 0.9 ^c^
	Agitation	32.3 ± 0.4 ^e^	12.9 ± 0.2 ^g^	17.8 ± 0.1 ^e^	22.1 ± 1.7 ^de^	52.9 ± 0.8 ^b^	83.7 ± 2.7 ^cd^
	Temporary immersion	26.3 ± 0.5 ^b^	9.4 ± 0.6 ^d^	11.0 ± 1.2 ^cd^	10.8 ± 2.1 ^abc^	46.6 ± 0.8 ^ab^	55.9 ± 2.4 ^b^

**Table 3 biomolecules-14-01132-t003:** Accumulation of flavonoids and 1,4-naphtoquinones (mg × 100 g^−1^ DW) in *D. peltata*, *D. indica*, *D. regia*, and *D. binata* cultivated in solid medium, agitated cultures, and temporary immersion bioreactor Plantform^TM^. Different letters indicate statistical significance of means according to two-way ANOVA, post hoc Tukey test at *p* < 0.05; SD—standard deviation, DW—dry weight, nd—not detected.

		Hyperoside	Myricetin	Quercetin	Kaempferol	Plumbagin	Ramentaceone
		mg ×100 g^−1^ DW [±SD]					
*Drosera peltata*	Solid medium	127.3 ± 3.5 ^cd^	44.2 ± 0.7 ^abc^	136.9 ± 0.7 ^ab^	17.1 ± 0.3 ^a^	1548.1 ± 60.8 ^bc^	nd
	Agitation	170.5 ± 1.3 ^d^	47.0 ± 2.5 ^abc^	102.5 ± 76.0 ^a^	50.7 ± 1.2 ^b^	1186.2 ± 40.2 ^a^	nd
	Temporary immersion	47.6 ± 0.2 ^a^	35.2 ± 0.4 ^ab^	72.4 ± 1.0 ^a^	13.5 ± 0.4 ^a^	1492.5 ± 43.0 ^abc^	nd
*Drosera indica*	Solid medium	55.6 ± 0.6 ^ab^	77.9 ± 8.9 ^abc^	94.5 ± 1.4 ^a^	11.4 ± 0.5 ^a^	1448.0 ± 15.8 ^ab^	419.2 ± 6.1 ^a^
	Agitation	79.1 ± 0.3 ^abc^	44.7 ± 1.8 ^abc^	91.3 ± 3.6 ^a^	19.6 ± 0.1 ^a^	1769.4 ± 39.9 ^bcd^	752.4 ± 3.9 ^d^
	Temporary immersion	97.9 ± 0.3 ^bc^	20.6 ± 0.7 ^a^	189.9 ± 0.8 ^bc^	11.1 ± 0.8 ^a^	1812.1 ± 83.8 ^cd^	462.6 ± 19.9 ^b^
*Drosera regia*	Solid medium	1337.3 ± 29.6 ^i^	338.3 ± 38.4 ^f^	260.9 ± 34.6 ^c^	51.0 ± 13.4 ^b^	4069.7 ± 116.3 ^g^	1358.7 ± 36.9 ^e^
	Agitation	939.7 ± 27.7 ^g^	259.4 ± 9.4 ^e^	453.7 ± 26.6 ^d^	141.8 ± 15.5 ^d^	1924.5 ± 29.7 ^d^	653.8 ± 15.0 ^c^
	Temporary immersion	1150.3 ± 33.5 ^h^	331.6 ± 57.4 ^f^	233.51 ± 2.5 ^c^	83.3 ± 1.0 ^c^	4606.2 ± 39.3 ^h^	1268.0 ± 10.9 ^e^
*Drosera binata*	Solid medium	592.3 ± 19.3 ^f^	111.2 ± 1.8 ^cd^	982.0 ± 2.9 ^e^	96.9 ± 9.1 ^c^	2894.7 ± 165.7 ^e^	nd
	Agitation	579.5 ± 3.7 ^f^	147.7 ± 36.6 ^d^	908.3 ± 16.4 ^e^	123.0 ± 1.8 ^d^	3256.9 ± 178.1 ^f^	nd
	Temporary immersion	498.2 ± 14.9 ^e^	99.4 ± 2.9 ^bcd^	949.5 ± 3.9 ^e^	53.5 ± 2.2 ^b^	3052.5 ± 260.9 ^ef^	nd

**Table 4 biomolecules-14-01132-t004:** Minimal bactericidal concentration (µg DW × mL^−1^) of *D. peltata*, *D. indica*, *D. regia*, and *D. binata* tissues cultivated in solid medium, agitated cultures, and temporary immersion bioreactor Plantform^TM^ against *Staphylococcus aureus* ATCC 25923 and *Escherichia coli* ATCC 25922. DW—dry weight.

		*Staphylococcus aureus* ATCC 25923	*Escherichia coli* ATCC 25922
		[µg DW × mL^−1^]	
*Drosera peltata*	Solid medium	834	4168
	Agitation	834	2000
	Temporary immersion	834	4168
*Drosera indica*	Solid medium	417	1000
	Agitation	417	2000
	Temporary immersion	417	1000
*Drosera regia*	Solid medium	417	2000
	Agitation	417	2000
	Temporary immersion	417	2000
*Drosera binata*	Solid medium	417	4167
	Agitation	834	4167
	Temporary immersion	834	4167

## Data Availability

Raw data are available upon request from the corresponding authors.

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
