# Peer review of "Effect of Agitation and Temporary Immersion on Growth and Synthesis of Antibacterial Phenolic Compounds in Genus Drosera"

_biomolecules, 2024, doi:10.3390/biom14091132_

Round 1

Reviewer 1 Report

Comments and Suggestions for Authors

I made some recommendations on the text.

Author Response

Dear Reviewer 1,

We are very grateful for all the comments.

Please find attached the file with our responses.

Kind regards, 

Wojciech Makowski

Reviewer 2 Report

Comments and Suggestions for Authors

The text bellow contains comments on manuscript entitled “Effect of agitation and temporary immersion on growth and synthesis of antibacterial phenolic compounds in genus Drosera”

I am listing some comments that the authors might take into consideration in case they find them useful:

To my opinion, the working principle of the temporary immersion systems should be also explained along with mentioning their advantages.

Considering that you present “the sum of total phenolic compounds”, I would suggest that you also measure the total phenolics according to the Folin-Ciocalteu method.

In case you have performed NMR analysis you should describe the sample preparation and condition of analysis. Also an NMR spectrum in the main text or as supplementary should be presented.

Comments on the Quality of English Language

Minor editing of English language required.

Author Response

Dear Reviewer 2,

we are very grateful for all the comments.

Please find attached the file with our responses.

Kind regards,

Wojciech Makowski
